# A systematic review of causes of recent increases in ages of labor market exit in OECD countries

**Michaël Boissonneault** [1]*, **Jaap Oude Mulders**[1], **Konrad Turek**[1,2], **Yves Carriere**[3]

**1** Netherlands Interdisciplinary Demographic Institute (NIDI-KNAW) / University of Groningen, The Hague, The Netherlands, **2** Department of Sociology, University of Amsterdam, Amsterdam, The Netherlands, **3** Département de Démographie, Université de Montréal, Pavillon Lionel-Groulx, Montréal, QC, Canada

* boissonneault@nidi.nl

**Data Availability Statement:** Systematic review. Data were extracted from research articles available via publishers of scientific journals. Data on article searches are available within the manuscript and its Supporting Information files.

## Abstract

Ages of labor market exit have increased steadily since the late 1990s in OECD countries, but with continuing population aging, there are calls for further stimulation of labor force participation at older ages. Social scientists have extensively studied causes of variation in retirement timing between individuals and across countries, but have paid less attention to causes of variation over time. This study systematically reviews evidence of causes of increases in ages of labor market exit over the past 30 years in OECD countries. Two goals are pursued: first, to provide an overview of the retirement domains that have been subject to investigation; second to compare studies with respect to the magnitude of change in retirement behavior that they attributed to different causes, in different contexts. Nineteen studies were reviewed. Available evidence articulates itself around four domains: inter-cohort changes in labor force participation of women (3 studies), educational attainment (3 studies) and lifetime wealth (1 study), and changes to social security systems (16 studies). Determinants in all domains explain a significant amount of past increases in ages of labor market exit, though figures attributable to similar determinants vary between studies and across countries. Evidence suggests that further postponement of labor market exit may depend on further increases to normal retirement ages and more limited access to early retirement programs, but also on further increases in educational attainment and the continued integration of women in the labor market. However, a large share of the past increases in ages of labor market exit remains unexplained; therefore, other factors such as those related to work and organizational characteristics deserve further research.

## Introduction

Since the late 1990s in Organization for Economic Co-operation and Development (OECD) countries, labor force participation at older ages has increased steadily. While less than half of the adults aged 55 to 64 were active in the labor market in 1996, nearly two-thirds of them were active in 2016 [1] (Fig 1). This constitutes a reversal of the historical trend, as labor force

**Funding:** KT was supported by the European Union's Horizon 2020 research and innovation programme under the Marie Skłodowska-Curie grant agreement No 748671 – LEEP – H2020-MSCA-IF-2016/H2020-MSCA-IF-2016. This work was supported by the Koninklijke Nederlandse Akademie van Wetenschappen (KNAW) and the University of Groningen.

**Competing interests:** The authors have declared that no competing interests exist.

participation at older ages had previously always been constant or declining [2]. Now, for the first time in history, each younger birth cohort can expect to exit the labor market at a later age than previous cohorts. For example, an average worker in 1996 could expect to exit at approximately age 62, whereas the same worker in 2016 could expect to exit at age 64 [3]. Increases in ages of labor market exit are pervasive, since they have occurred in all OECD countries (despite variation in timing of onset and magnitude), and they have affected both men and women of all socio-economic classes[4].

These developments have caught the attention of researchers in social sciences and recent studies have investigated the causes for the increases in ages of labor market exit in OECD countries. Synthesizing the increasing amount of evidence on what causes increases in ages of labor market exit over time could prove highly valuable in the context of ongoing population aging[5]. As the proportion of older people in the population increases and the proportion of younger people stagnates or declines, scientists and policy makers alike are calling for further stimulation of labor force participation at older ages[6,7]. Therefore, we aim to systematically review the available scientific evidence of the causes for increases in ages of labor market exit in OECD countries in recent decades.

Reviews have summarized the evidence on the causes for variation in retirement timing between individuals[8–10]. Results showcase high agreement on the determinants of such variation; these are summarized in Table 1 referring to four retirement domains (individual, job, family, and socio-economic). Information contained in this table will serve as a reference for the remainder of this article.

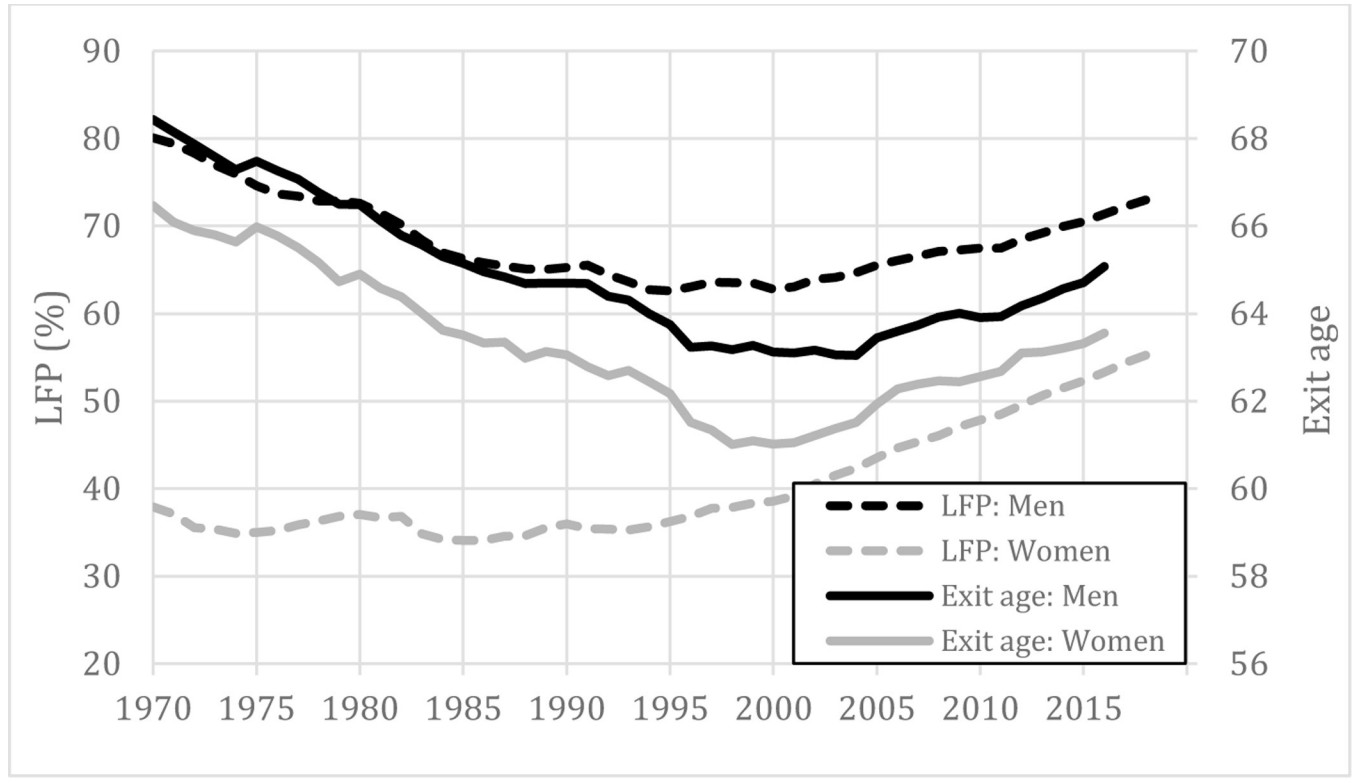

**Fig 1. Labor force participation at ages 55 to 64 (left axis) and average labor market exit age (right axis) for OECD countries.** Labor force participation rates are calculated as the total of people in the labor force divided by the total population ages 55 to 64. [1] The average effective age of labor market exit is based on changes in labor force participation rates and are therefore not affected by the proportion of people working. [3].

**Table 1. Overview of retirement domains to which current evidence on causes of variation between individuals in ages of labor market exit belong.** Adapted from Wang & Schultz 2010, Fisher, Chaffee & Sonnega 2016 and Scharn et al. 2018.

| INDIVIDUAL | JOB | FAMILY | SOCIO-ECONOMIC |
|---|---|---|---|
| • Demographic characteristics (age, education)<br>• Personality, needs, motivations and values<br>• Knowledge, skills and abilities<br>• Attitudes towards work and retirement<br>• Health and lifestyle<br>• Employment history<br>• Income, wealth and health insurance | • Job characteristics<br>• Age stereotypes and norms, diversity and discrimination<br>• HR policies and practices<br>• Employer provided pension plan<br>• Training/skill development opportunities | • Caregiving responsibilities<br>• Partnership status and relationship's quality<br>• Partner's retirement status | • Social norms about retirement<br>• Macroeconomic conditions<br>• Social security systems |

Another important strand of literature investigated causes for variation in retirement behavior between countries at one point in time [11–13]. This work concentrated on the incentives created by social security in inducing retirement at specific ages and developed the concept of implicit tax on work. The finding that a higher implicit tax on work correlates with lower retirement ages prompted countries to introduce changes to their social security systems with the aim of encouraging later retirement [3]. These changes likely played a role in the recent increases in ages of labor market exit and are summarized in Table 2 [11].

Though causes for differences in retirement timing between individuals as well as between countries were already reviewed, we are not aware of any study that reviewed causes for changes in retirement behavior *over time*. By filling this gap, this systematic review contributes to the current state of knowledge in two main ways. Firstly, it summarizes and synthesizes the available evidence on factors that affect changes in ages of labor market exit over time. As a result, it also identifies knowledge gaps in our understanding of what causes increasing ages of labor market exit, thereby providing guidance for future research on this topic. Secondly, it compares studies with respect to the magnitude of change in retirement behavior that was attributed to different causes, in different contexts. Effects are presented individually for each cause, referring to the context in which they were estimated. Assessing whether systematic differences emerge between causes and the contexts in which they were studied may prove instrumental in informing policy on how to further increase ages of labor market exit in the context of population aging.

We reviewed articles that aimed at explaining why ages of labor market exit have increased in the last decades in OECD countries. In the studies reviewed, changes in the age of labor market exit were measured in the form of either changing rates of labor force participation (LFP) or changing retirement probabilities. LFP in older age groups captures both employment and retirement patterns within the group. Retirement probabilities, on the other hand, are obtained by following working individuals over a period and recording retirement

**Table 2. Classes of changes brought to social security systems over the last decades in OECD countries.** Adapted from Börsch-Supan & Coile 2018.

| |
|---|
| • Change to retirement age or in years of contribution required [early or normal] |
| • Change to programs allowing partial retirement |
| • Change to the generosity of social security benefits |
| • Change to the actuarial adjustment of social security benefits [early or delayed claiming] |
| • Change to earnings tests |
| • Change to pension plans [e.g. defined benefits to defined contribution] |
| • Change to early retirement, disability insurance and unemployment insurance programs |

occurrences. In comparison to changes in LFP, changes in retirement probabilities over time reflect variation in retirement behavior more closely. For ease of interpretation, working life expectancies are often calculated from retirement probabilities, referred to as the effective retirement age (ERA). ERA can be contrasted with the normal retirement age (NRA), which is set by law and determines the age at which full pension benefits are granted. In the studies reviewed here, changes in retirement behavior were most of the time measured in the form of changes in LFP or ERA. In the remainder of this article, changes in retirement behavior will thus be referred to as changes in LFP/ERA.

We first identify domains for which evidence is available regarding changes in LFP/ERA. Then, we present results regarding the amount of change that is attributable to each domain, as estimated in each study and by country and gender, if applicable. Studies were divided into two groups according to the approach that was taken for explaining change over time in LFP/ERA: the first group considers differences in LFP/ERA between two points in time, while the second one considers differences between groups that were differently affected by external factors such as pension reforms. Results that were extracted from the first group of studies are proportions of change in LFP explained by a specific factor as well as total change in LFP during the period under study, while results that were extracted from the second group of studies are regression outputs (e.g., coefficients) that give the ceteris paribus effect of an exogenous change on retirement behavior. This review is limited to OECD countries, which share similarities regarding social security systems, population structure, and trends in age of labor market exit. This review is further limited to studies that address national populations or population subgroups (e.g., men of a specific age range during a period), and thus studies of narrow samples (e.g., professional groups) were excluded.

## Materials and methods

### Database search

We systematically searched databases EconLit, PubMed, and Web of Science. The same search terms, adjusted for syntax requirements, were used in each database. Four strings of words were identified and combined to use during a single search within titles. The first string contained the words *raise*, *labor force participation*, and *old age*, the second *raise* and *retirement age*, the third *extension* and *working life*, and the fourth *delay* and *retirement*. The terms that formed each string were combined using the Boolean operator AND, and equivalent expressions were used where applicable using OR (e.g., *delay* OR *postponement of retirement*) (S2 Table).

### Inclusion criteria

We included peer-reviewed research articles, published in English since the year 2000, which analyzed the general population of OECD countries (excluding studies of specific subgroups, e.g. occupational groups). Studies were included when they assessed retirement behavior (not retirement intentions), and aimed to explain changes in LFP/ERA over time (not between individuals or across countries). Research designs must have included quantifiable changes in retirement behavior (e.g., proportion employed, effective retirement age) as outcome variables over well-defined periods, or birth-cohorts, and age-groups, while explanatory variables must have denoted changes over time in any explanatory factor. No criteria were applied concerning the longitudinal or cross-sectional design of studies.

## Selection procedure

Searches were conducted simultaneously and independently by three researchers (i.e., J.O.M., K.T., and M.B.) on 13 February 2019. Duplicates were removed. Screening was performed in three rounds: first based on titles, then on abstracts and to finish on full texts. Criteria regarding article formats and populations were applied throughout; those regarding a study's purpose were applied from the second round forward, and those regarding research designs during the last round only. During each round, two of the three researchers (i.e., J.O.M., K.T., and M.B.) reviewed studies independently. In case of disagreement regarding inclusion, the third researcher made the final decision. Additional articles were considered based on reference lists of the studies that passed the second round of screening as well as on expert knowledge (Fig 2).

## Data extraction

Data regarding a study's design, results and methodology were retrieved manually from each article. Information on study design included population of interest (e.g. married men), country, age and year ranges, the dependent and main independent variable(s) and their measurement (e.g. percentage points, year, probabilities). Information on results included, if available, changes due to a cause of interest in labor force participation rates, retirement ages or retirement probabilities, or otherwise coefficient values in regression output. Values were extracted concerning each predictor of interest, separately for men and women and each country, if applicable. Values pertaining to the preferred specification were extracted, if indicated, or averaged over the different specifications otherwise. Information on methodology included the dataset(s) used, number of observations, the statistical model used, and the strategy adopted for tracking causality. To facilitate interpretation, we consider two main classes of results. The first one includes results from studies designed to explain differences in LFP between two points in time. Since the distance between these points in time may vary considerably, results are presented in terms of yearly change in LFP. The second class includes results from studies designed to explain differences in retirement behavior between two groups that were affected differently by external factors such as pension reforms. This class is further subdivided into four subclasses denoting different outcome variables: LFP rates, retirement ages, retirement probabilities and hours worked. Articles were deemed as having made effort towards tracking causality if they used statistical models controlling for potentially spurious correlation and included instruments controlling for endogenous relations, for example by using control groups not affected by the independent variable of interest. The PRISMA statement checklist was referred for the review process (S3 Table).

## Results

Six hundred eighty-nine studies were identified through database searches, of which 511 remained after duplicates were removed. Studies were then considered for inclusion based on selection criteria (see Materials and Methods). Information included in article titles allowed us to exclude 386 studies, and information in abstracts and texts allowed us to exclude another 108. Following this procedure, 17 studies were selected for assessment. The reference lists of these studies were checked, and experts in the field were consulted regarding missing studies, which added 31 more. Following assessment of these studies, 19 were included in the final selection (Fig 2).

Articles concentrated on 11 countries, all of which are located in Europe or North America. Most assessed single countries, but two included cross-national comparisons. They focused most often on the United States (6 studies), followed by Germany (5 studies). Austria, Sweden,

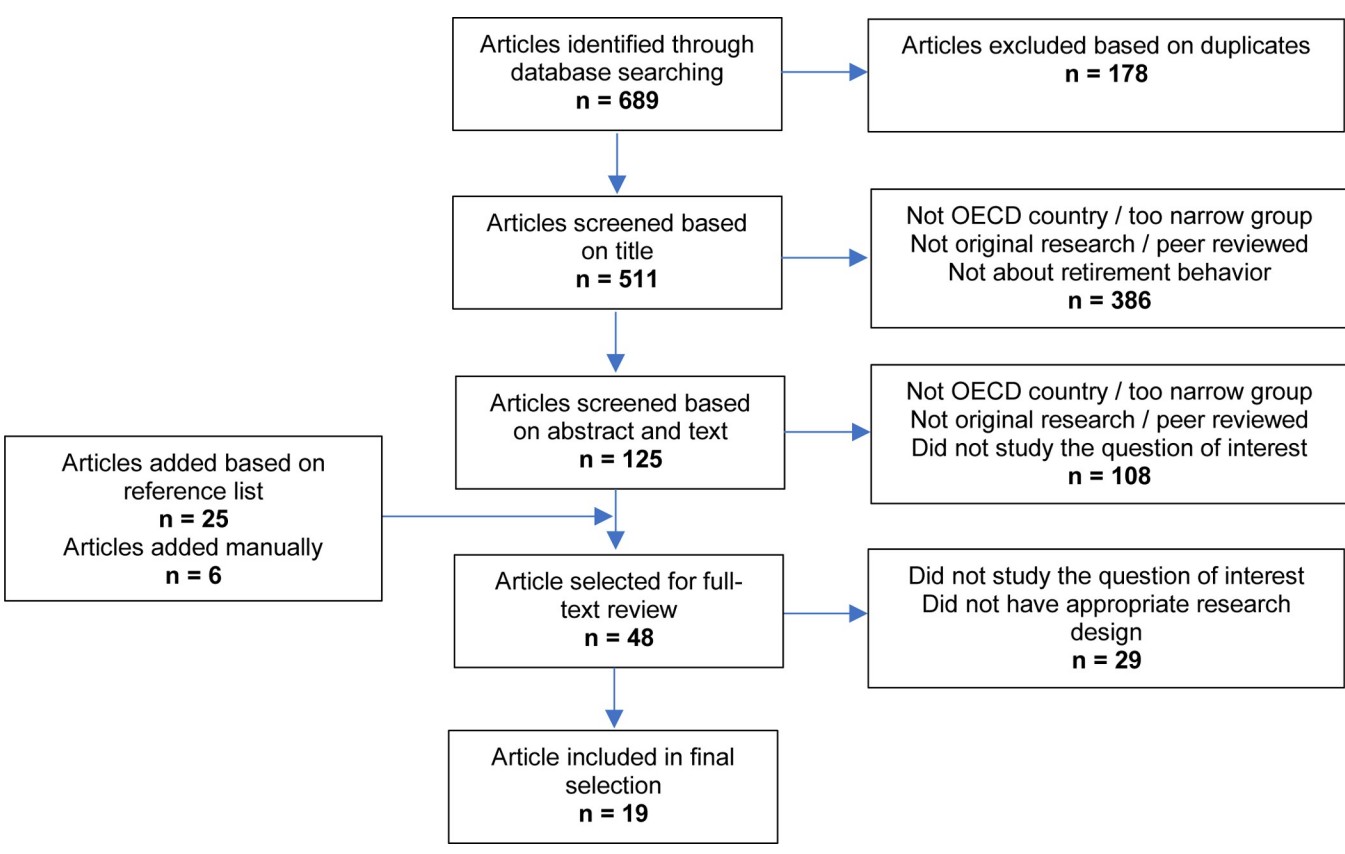

**Fig 2. Decision tree for article inclusion.**

and the United Kingdom were studied two times each, and Belgium, Canada, Denmark, Estonia, Spain and Switzerland were studied once. Studies covered periods of varying lengths. The shortest period covered one and a half years (2007–2008.5) while the longest covered 17 years (1988–2005). Other studies followed birth-cohorts. The oldest cohort was born in 1928 and the youngest in 1951. Outcomes were measured among women slightly less than half of the time, and the studies considered varying age ranges. Some considered changes in retirement behavior in a narrow age group (e.g., 62 to 63), while others considered groups of greater width (e.g., up to 20 years). Most studies concentrated on age groups strictly before or both before and after the NRA, while few considered age groups strictly after the NRA. Studies analyzed data from national registers, labor force surveys, and surveys that were representative of national populations (e.g., Health and Retirement Study). Excepting two [12,13], all articles assessed causal relationships between explanatory variables and an outcome (S1 Table).

## Domain and outcome coverage

Table 3 shows studies classified regarding the explanatory and outcome variables contained in their analyses. Explanatory variables covered 4 of the 18 domains identified in Table 1, including partner's retirement status (3 studies), demographics (3 studies), income, wealth, and health insurance (1 study), and social security systems (16 studies). Studies that investigated changes to social security systems were further broken down according to the categories identified in Table 2. Seven studies considered the effects of increases in LFP/ERA in terms of

Table 3. Domain and outcome coverage among reviewed studies.

| Domains | Type of outcome | | | | No. of results |
|---|---|---|---|---|---|
| | Labor force participation rates | Effective retirement age | Retirement probabilities | Hours worked | |
| **Partner's retirement status** | Blau & Goodstein (2010) Pérez et al. (2020) Schirle (2008) | | | | 3 |
| **Demographic characteristics (education)** | Blau & Goodstein (2010) Larsen & Pedersen (2017) Schirle (2008) | | | | 3 |
| **Income, wealth and health insurance** | Blau & Goodstein (2010) | | | | 1 |
| **Social security systems, including:** | | | | | 16 |
| *Retirement age / years of contribution (early or normal)* | Blau & Goodstein (2010) Dejemeppe et al. (2015) Gustman & Steinmeier (2009) | Mastrobuoni (2009) Puur et al. (2015) | | | 5 |
| *Partial retirement programs* | Dejemeppe et al. (2015) | | | | 1 |
| *Benefits' generosity* | Dejemeppe et al. (2015) Staubli & Zweimüller (2013) | | Hanel & Riphahn (2012) | | 3 |
| *Actuarial adjustment of benefits* | Blau & Goodstein (2010) Gustman & Steinmeier (2009) | Berkel & Börsch-Supan (2004) | Buchholz et al. (2013) | | 4 |
| *Earnings tests* | Gustman & Steinmeier (2009) | | | Disney & Smith (2002) | 2 |
| *Pension plan* | Hurd & Rohwedder (2011) | Friedberg & Webb (2005) Qi et al. (2018) | Buchholz et al. (2013) | | 4 |
| *Early retirement, disability insurance and unemployment insurance programs* | Dejemeppe et al. (2015) Staubli & Zweimüller (2013) Staubli (2011) Hanel (2010) | Berkel & Börsch-Supan (2004) Bönke et al. (2018) | Buchholz et al. (2013) | | 7 |
| **No. of results** | 10 | 6 | 2 | 1 | |

changes to early retirement and related programs. Five considered changes to statutory retirement ages or the number of years of required contribution for early or normal retirement. Four investigated changes to pension plans over time (i.e., defined benefits to defined contributions) and the same number considered changes to actuarial adjustments of benefits or earnings tests. Three studies estimated the effects of changes to benefits generosity over time, usually in the direction of less generous benefits (two of three studies). Two studies estimated the effect of removing earnings tests, and one investigated changes to regulations regarding partial retirement. Some studies examined the combined effect of multiple changes to social security systems and therefore appear in different categories (to be hereafter reffered to as "extensive" reforms). Regarding outcome variables, changes in retirement behavior were measured ten times in terms of LFP, six times in terms of ERA, two times in terms of retirement probabilities and one time in terms of hours worked.

## Differences in LFP/ERA between two points in time

Five studies considered differences in LFP between two points in time and aimed at explaining it in terms of change in one or more independent variables. Table 4 presents an overview of these studies' designs while Fig 3 presents the amount of yearly change in LFP that was observed (full bars), broken down by the amount that was explained by each variable of interest (lower part), and the amount that was not explained by each variable of interest (upper part). Levels concern the mean annual change over the whole period of observation to improve comparability as year ranges vary. In total, results were available for 20 effects covering eight countries and six classes of predictors. These predictors included changes in inter-cohort educational attainment, LFP of women, normal retirement age, delayed retirement credits, lifetime earnings, and extensive social security reforms. The proportion of change in LFP

**Table 4. Overview of the first group of studies which investigated differences in LFP between two points in time.**

| Author | Year | Countries | Subpopulation | Age range | Year range | Effect(s) studied | Details | Causal |
|---|---|---|---|---|---|---|---|---|
| Blau | 2010 | United States | All men | 55–69 | 1988–2005 | Delayed retirement credits (DRC) | Introduction of credits for delayed retirement past the NRA over the period 1987 to 2005 | Yes |
| | | | | | | NRA | Increase of normal retirement age from age 65 to 65.5 | |
| | | | | | | Lifetime earnings (LE) | Increases in total lifetime earnings | |
| | | | | | | LFP women | Increases in labor force participation of women | |
| | | | | | | Educational attainment | Increases in inter-cohort educational attainment | |
| Dejemeppe | 2015 | Belgium | Initially employed men and women | 50–59 | 2004–2013 | Extensive reform (ER) | Reduction in employers' social security contributions for workers aged 50–56 | Yes |
| | | | | | | | Stricter admissibility criteria to early retirement | |
| | | | | | | | Higher age of admissibility to early retirement | |
| | | | | | | | Easier access to partial retirement | |
| | | | | | | | Increase in the generosity of retirement benefits | |
| Larsen | 2017 | Denmark, Germany & Sweden | All men and women | 65–69 | 2004–2013 | Educational attainment | Increases in inter-cohort educational attainment | No |
| Pérez | 2020 | Spain | All men who live with a partner | 55–64 | 1995–2016 | LFP women | Increases in labor force participation of women | Yes |
| Schirle | 2008 | Canada, United Kingdom & United States | All married men | 55–64 | 1994–2005 | Educational attainment | Increases in inter-cohort educational attainment | Yes |
| | | | | | | LFP women | Increases in labor force participation of women | |

LFP=Labor force participation rate; MR=Multiple reforms; NRA=Normal retirement age; DRC=Delayed retirement credits; LE=Lifetime earnings. M=Men; W=Women.

attributable to different predictors varies strongly between studies. The effect of educational attainment varies from 0.004 pp per year (women in Germany in [13]) to 0.21 pp per year (men in the United States in [14]). Likewise, the proportion of change in LFP attributable to change in LFP of women varies from 0.04 pp per year (men in the United States in [15]) to 0.43 pp per year (men in Spain in [16]).

## Differences in LFP/ERA between groups

Fourteen studies investigated differences in LFP/ERA between groups. These aimed at identifying the ceteris paribus effect of an exogenous change on retirement behavior, for example brought about by a reform of the social security system. Table 5 presents an overview of the studies' designs while their results are illustrated in Fig 4, where each panel refers to a different type of outcome (LFP, ERA, retirement probabilities and hours worked). Six types of reforms in social security systems are considered: extensive pension reform, pension plan, disability benefits, early retirement age, earnings tests, benefits reduction and NRA. Differences among subgroups with regards to the way that they were affected by reforms of social security systems vary greatly, stretching from nearly null effects (men and women in Germany in[17]; women in Germany in[18]; women in Sweden in[19]) to considerable ones, sometimes above 20 pp

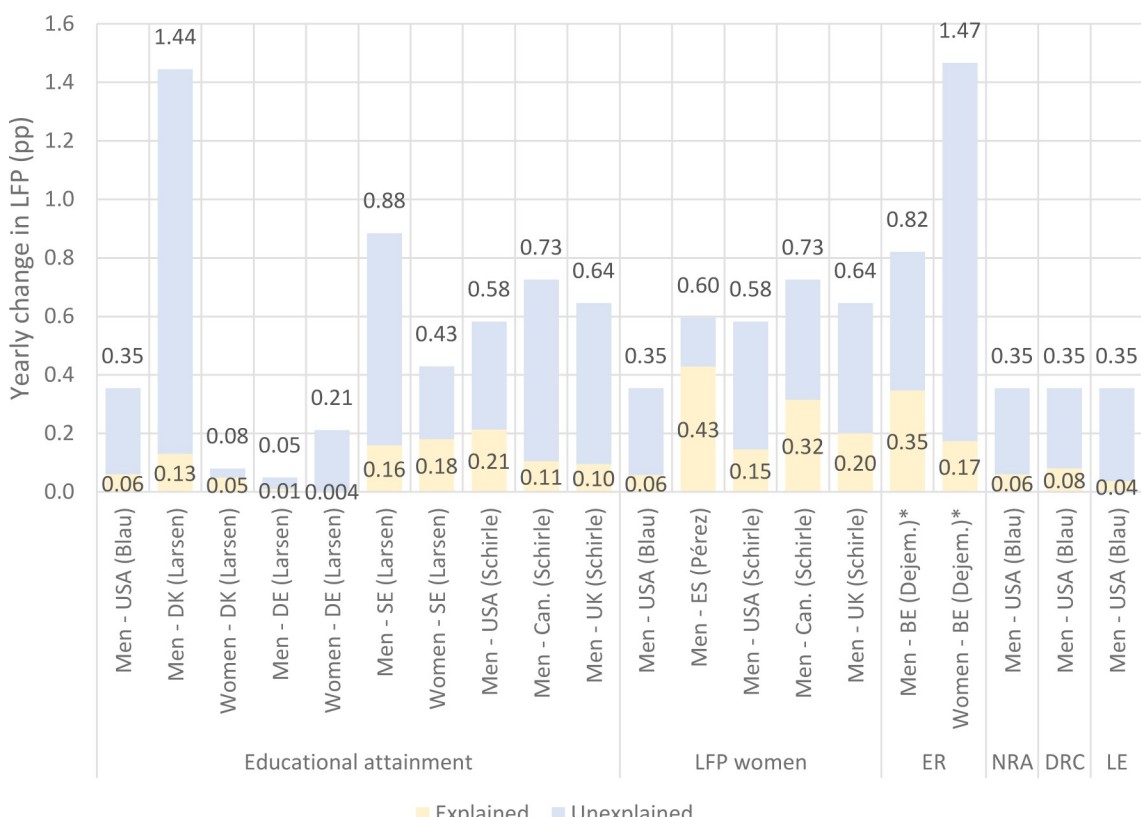

**Fig 3. Differences in LFP between two points in time and proportion attributable to specific factors.** Figures correspond to the yearly change in LFP (or LFP equivalent), by gender and country. Results are grouped according to the explanatory variables that are indicated at the bottom of the graph. Articles from which results were retrieved are referred to by the first author's name (in parentheses). Numbers above bars indicate the total observed change; numbers below refer to the explained part. For example, in a study by Larsen among men in Denmark, the observed yearly change in LFP was 1.44 pp, of which 0.13 pp was explained by increases in educational attainment. Details about the studies' designs are provided in Table 4. Details about the calculations made are presented in S1 Table. LFP=Labor force participation rate; MR=Multiple reforms; NRA=Normal retirement age; DRC=Delayed retirement credits; LE=Lifetime earnings. M=Men; W=Women. * The authors note a lack of statistical power to draw firm conclusions.

(pension reform affecting men and women in Austria in[20]; the removal of earnings tests for men in the UK in[21]). Once again, there does not seem to be one particular strategy which consistently brings about similar changes in LFP/ERA, or one particular context that saw stronger increases.

## Discussion

The majority of the reviewed articles considered changes to social security systems as a cause for the recent increases in LFP/ERA. Extensive reforms in Germany and Austria induced strong increases in LFP/ERA over periods ranging between seven and twenty years [18,20,22], though effects tended to be weaker among women in Germany [18,22]. A key element of the reforms in both countries was the introduction of financial penalties for claiming benefits before the NRA. In the United States, changes to social security introduced in the 1990s and 2000s also contributed to increases in LFP/ERA. The combined effect of the introduction of delayed retirement credits, the removal of earnings tests, and increases to the NRA brought about a 2 pp increase in LFP among men ages 65–67 [23], while the increase in NRA induced increases in LFP of 0.05 pp per year among men ages 55–69 [15] or a postponement of

**Table 5. Overview of the second group of studies which investigated differences in LFP/ERA between age- or cohort-groups, grouped by type of outcome studied.**

| Author | Year | Countries | Subpopulation | Age range | Year range | Effect(s) studied | Details | Causal |
|---|---|---|---|---|---|---|---|---|
| *Outcome*: Labor force participation rates | | | | | | | | |
| Gustman | 2009 | United States | Married men | 65–67 | 1992–2004 | Extensive reform (ER) | Increase of normal retirement age from age 65 to 65.17 | Yes |
| | | | | | | | Introduction of credits for delayed retirement past the NRA over the period 1987 to 2005 | |
| | | | | | | | Reduction (1990) and elimination (2000) of the implicit tax on earnings past the normal retirement age | |
| Hurd | 2011 | United States | Initially employed men and women | 61–68 | 1992–2004 | Pension plan | Decrease in the proportion of workers with pension plan with defined benefits and increase in the proportion with pension plan with defined contribution | Yes |
| Staubli | 2011 | Austria | Men and women working in private sector | 55–56 | 1994–1999 | Disability benefits | Increase of availability to disability benefits from age 55 to 57 | Yes |
| Staubli & Zweimüller | 2013 | Austria | Men and women working in private sector | 57–64 (men) and 52–59 (women) | 2000–2010 | Early retirement age | Increase of early retirement age from age 60 to 62 (men) and age 55 to 58.2 (women) | Yes |
| *Outcome*: Hours worked per week | | | | | | | | |
| Disney | 2002 | Great Britain | All men and women | 60–74 (men) and 55–69 (women) | 1986–1994 | Earnings tests | Abolition of the earnings tests in 1989 | Yes |
| *Outcome*: Retirement ages | | | | | | | | |
| Berkel | 2004 | Germany | All men and women | 55–70 | 1984–1997 | Extensive reform (ER) | 0.3% benefit reduction per month of early retirement | Yes |
| | | | | | | | 0.5% pension increase per month of work past the normal retirement age | |
| | | | | | | | Restricted access to disability pension | |
| Bönke | 2018 | Germany | All men | 63–65 | 2004–2012 | Benefit reduction | 0.3% benefit reduction per month of early retirement | Yes |
| Friedberg | 2005 | United States | Men and women initially in full employment | 63–65 | 1983–2015 | Pension plan | Decrease in the proportion of workers with pension plan with defined benefits and increase in the proportion with pension plan with defined contribution | Yes |
| Hanel | 2010 | Germany | Initially employed men and women | 55–67 | 1995–2002 | Extensive reform (ER) | 0.3% benefit reduction per month of early retirement | Yes |
| | | | | | | | Gradual increase in early retirement ages over period 1997–2005 | |
| Mastrobuoni | 2009 | United States | All men and women | 62–65 | 1989–2007 | NRA | Increase of normal retirement age from age 65 to 65.67 | Yes |
| Puur | 2015 | Estonia | All women | unspecified | 2002–2011 | NRA | Increase of normal retirement age from age 58.5 to 61.5 | No |
| Qi | 2018 | Sweden | Initially employed men and women | 60–67 | 1997–2011 | Pension plan | Shift in pension plans from defined benefits to notional defined contribution | Yes |
| *Outcome*: Retirement probabilities | | | | | | | | |
| Buchholz | 2013 | Germany | Initially employed men and women | 60–70 | 1984–2007 | Extensive reform (ER) | 0.3% benefit reduction per month of early retirement | Yes |
| | | | | | | | 0.5% pension increase per month of work past the normal retirement age | |
| | | | | | | | | |
| Hanel & Riphahn | 2012 | Switzerland | Initially employed or unemployed men and women | 62–64 | 2000–2005 | Benefit reduction | 3.4% reduction age 62 year 2000 | Yes |
| | | | | | | | 6.8% reduction age 62 year 2001–2004 | |
| | | | | | | | 3.4% reduction age 63 year 2005 | |

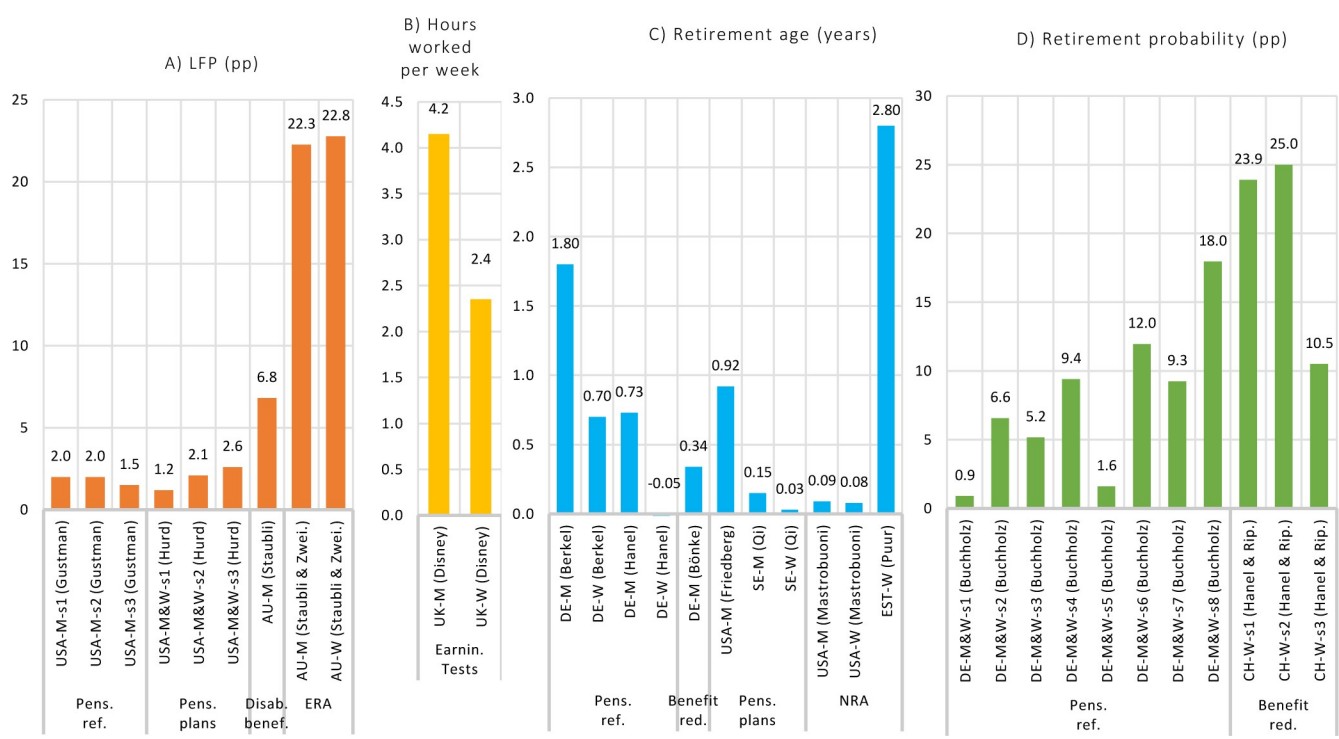

**Fig 4. Differences in LFP/ERA between groups attributable to specific explanatory variables, by type of outcome (panels A-D), gender, specification and country.** Results are grouped in panels according to outcome measures. They are further grouped by the explanatory variables indicated at the bottom of the graph. Articles from which results were retrieved are referred to by the first author's name (between parentheses). For example, in a study by Berkel among men in Germany, the effect of treatment (pension reform) measured as a difference in retirement age between treatment and control group was 1.8 year. Further details about the studies' designs are provided in Table 5. Details about how the figures were extracted from the articles are available in the appendix. Pens. ref=Extensive pension reform; Pens. plans= Change in pension plans; Disab. benef.= Restricted disability benefits; ERA= Increased early retirement age; Earnin. tests= Removal of earnings tests; Benefit red.= Benefit reduction; NRA = Increase of NRA. M=Men; W=Women.

retirement of 0.5 years per year of increase in NRA among those ages 62–65 [24]. The phasing in of delayed retirement credits was responsible for a 0.06 pp increase per year in LFP among American men ages 54–69 [15].

The decrease in the proportion of workers having defined benefits pension plans and the increase in the proportion of those having defined contribution plans in the United States (or with a notional defined contribution in Sweden) caused considerable increases in LFP among men, but weaker ones among women [19,25,26]. The removal of earnings tests past the NRA induced strong increases in the number of hours of work among men ages 65–69 and women ages 60–64 in the United Kingdom [21]. Other changes to social security systems included a reduction of benefits generosity linked to early retirement among German men [27] and Swiss women [18] and the tightening of admissibility criteria to disability benefits among Austrian men [28]. These interventions were more targeted as they affected narrower age groups but induced clear changes in retirement behavior among these groups.

Changes to social security systems clearly induced prolonged labor force participation in recent decades, but they could not explain all recent increases in LFP/ERA. Studies that assessed increases in LFP in terms of changes to social security systems explained, at best, half of the increases [28], though most attributed 30% to 40% of the increases to such changes [15,19,20,29]. Four studies investigated the effect of factors not related to social security systems on increases in LFP/ERA. Changes in educational attainment across birth-cohorts of men increased LFP by 0.01 to 0.21 pp per year, depending on the country and the study [13–

15]. Only one study [13] considered changes in educational attainment over cohorts of women, finding a contribution of 0.004 to 0.18 pp over the 2004 to 2013 period, depending on the country. Increases in labor force attainment of women induced increases in LFP of married men that ranged from 0.04 pp per year in the United States [15] to 0.43 in Spain [16].

## Research gaps

One article combined the effects of changes to social security systems (i.e., higher NRA, actuarial adjustment, benefits generosity), changes in educational attainment of successive cohorts, and increases in labor force attainment of married women, explaining approximately 73% of the observed increases in LFP [15]. Other articles assessed fewer predictors, explaining much smaller proportions of the increases. There are other factors which are conceptually likely to have affected LFP in recent years that have yet to have been subject to empirical analyses (see Table 1). For example, despite ongoing increases in older adults' health and their ability to work [30,31], no study assessed the influence of this trend on ages of labor market exit. Similarly, individuals with more complex career paths tend to exit the labor market at higher ages [32] but it is not clear whether the de-standardization of careers has contributed to increased LFP at older ages [33]. Changes in organizations, such as improved accommodative HR practices for older workers and progress with counteracting age discrimination at work [34] might have contributed to increases in ages of labor market exit. Also, employers are more likely to employ older workers as they increasingly recognize their value and are getting more experienced with dealing with an aging workforce [35]. Decreasing physically demanding jobs, improved quality of work, and the rise of flexible work arrangements might also have facilitated higher ages of labor market exit [36]. Changing societal norms regarding work at later ages and retirement may have also contributed to increases in LFP, as younger cohorts have different attitudes, preferences, and expectations regarding work and retirement than older cohorts [37]. Finally, following the Great Recession of 2007–2008, studies could investigate factors like decreased coverage of private pension plans, higher debt load among older adults, and decreased returns on private savings which decreased disposable income and may have forced older adults to delay retirement [38].

## Future prospects

As of 2017, increases to the NRA were planned in close to half of OECD countries, including, in some countries, automatic links to changes in life expectancy [3]. Given the evidence reviewed in this paper, further increases in LFP/ERA are to be expected following these changes. In contrast, other types of changes to social security systems may become more difficult to implement. Reductions to the generosity of retirement benefits and tightening of admissibility criteria must be implemented carefully since they might negatively influence economic wellbeing at older ages. Other changes, such as those pertaining to actuarial treatment of pension benefits, changes from defined benefits to defined contribution schemes, and elimination of earnings tests, can be implemented only once, and thus countries that have already implemented such changes cannot benefit from them in the future. Finally, the effects of other relevant factors might be slowly diminishing. For example, future increases in educational attainment across birth-cohorts will be milder than those that have prevailed until now [39]. The labor force participation of women is approaching that of men in many countries, and when the labor force attainment of married women starts leveling, the effect on the retirement timing of men might begin to wear off. However, factors such as more accommodating HR practices for older workers, improving quality of work, and changing societal norms may start playing a larger role.

## Limitations

Some limitations inherent to this review should be mentioned. The studies reviewed here were performed in countries that differ in their level of education, health, occupational structure, age composition, or social security systems. These factors may interact with reforms or processes, leading eventually to different effects for LFP/ERA. Additionally, studies concerned different time frames, different populations (e.g. labor or marital status), as well as different age groups or birth cohorts. Because of this, a more quantitative treatment of the results (e.g. meta-analysis) was not feasible, and results should be interpreted bearing the particularities of each study in mind. Furthermore, though studies covered eleven countries, nine out of nineteen studies strictly focused on either the United States or Germany, and only 11 countries of the total of 36 OECD countries were covered. Therefore, our conclusions mainly apply to the countries covered by the reviewed studies, and care should be taken when extrapolating the findings to other OECD countries.

## Conclusions

The last 30 years provided social scientists with the opportunity to gain direct evidence on what causes individuals to postpone retirement. This systematic review shows that although several studies investigated causes of recent increases in ages of labor market exit, the variety of topics on which they concentrated remains limited. In countries for which evidence was available, increasing the NRA and limiting access to and the generosity of early retirement programs, among other changes to social security systems, contributed to higher ages of labor market exit and increasing labor force participation at older ages. In the same countries, changes in the patterns of labor force participation among married women and in educational attainment across birth cohorts seem to have played similar roles, though evidence is less robust. Policies that aim at increasing ages of labor market exit should thus consider modifying the incentives created by social security in inducing retirement at specific ages, but also promoting education and life-long learning, and facilitating the integration of female workers to the labor market. Other factors such as changes in population health, the substantive nature of work, the role of HR practices, and norms and attitudes towards work at older ages may have been equally powerful in increasing ages of labor market exit, but evidence on the role they played is missing. Increasing the scope of evidence to other potential causes of increases in ages of exit from the labor market, as well as to more countries, will provide scientific grounds for stimulating further increases in ages of labor market exit in OECD countries.

## Supporting information

**S1 Table. Studies overview: Data source and methods.**
(DOCX)

**S2 Table. PRISMA 2009 checklist.**
(DOCX)

**S3 Table. Description of calculations, results reported in Fig 3.**
(DOCX)

**S4 Table. Description of calculations, results reported in Fig 4.** Effects (i.e. regression coefficients) were extracted directly from articles where available and appear in column "Effect". Estimates for separate groups appear in the columns "Value"; in such cases, it is the difference between groups that appears under "Effect".
(DOCX)

**S1 Data. Full strings used in searches.**
(DOCX)

## Acknowledgments

We thank Anthony Quinquis for help during selection of the studies.

## Author Contributions

**Conceptualization:** Michaël Boissonneault.

**Data curation:** Michaël Boissonneault, Jaap Oude Mulders, Konrad Turek, Yves Carriere.

**Methodology:** Michaël Boissonneault, Jaap Oude Mulders, Konrad Turek.

**Writing – original draft:** Michaël Boissonneault.

**Writing – review & editing:** Michaël Boissonneault, Jaap Oude Mulders, Konrad Turek, Yves Carriere.

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
