## [Decision Letter · Decision Letter 0]

3 Dec 2019

PONE-D-19-28928

A systematic review of causes of recent increases in ages of labor market exit in OECD countries

PLOS ONE

Dear Dr. Boissonneault,

Thank you for submitting your manuscript to PLOS ONE. After careful consideration, we feel that it has merit but does not fully meet PLOS ONE’s publication criteria as it currently stands. Therefore, we invite you to submit a revised version of the manuscript that addresses the points raised during the review process.

1. The topic is relevant and hence worthy of investigation, currently the paper lacks some important issues that should be addressed in a future version 

2. The reviewers did not consider the review systematic since only 18 papers were reviewed and there is a large body of the literature on the topic that should be included. Also, it would be important to developed a more quantitative analysis. One strong suggestion was to use a meta-analysis to study the problem. 

3. Introduction should state more clearly the contribution of the paper 

4. The concluding section should state more practical policy conclusion 

5. The conclusions and discussion are based on a set of few heterogenous studies. Thus, some conclusions are reached only by looking at results and it is necessary to perform a more quantitative study. 

6. it is hard to justify ruling-out cross-sectional studies since they can contain equally-valid causal claims about the causes of labor-market choices, so I think the rationale for this needs to be more explicit. 

7. The paper should note that there is earlier research that examines the role of job characteristics using linked survey and register data. https://doi.org/10.1016/j.jeoa.2019.02.001

8. paper needs some revision - text and structure 

9. Please, see more detailed comments below.

We would appreciate receiving your revised manuscript by Jan 17 2020 11:59PM. To enhance the reproducibility of your results, we recommend that if applicable you deposit your laboratory protocols in protocols.io, where a protocol can be assigned its own identifier (DOI) such that it can be cited independently in the future. For instructions see: http://journals.plos.org/plosone/s/submission-guidelines#loc-laboratory-protocols

We look forward to receiving your revised manuscript.

Kind regards,

Bernardo Lanza Queiroz, Ph.D

Academic Editor

PLOS ONE

Journal Requirements:

1. Please include captions for your Supporting Information files at the end of your manuscript, and update any in-text citations to match accordingly. Please see our Supporting Information guidelines for more information: http://journals.plos.org/plosone/s/supporting-information.

Reviewers' comments:

Reviewer's Responses to Questions

**Comments to the Author**

1. Is the manuscript technically sound, and do the data support the conclusions?

Reviewer #1: Yes

Reviewer #2: No

Reviewer #3: Partly

2. Has the statistical analysis been performed appropriately and rigorously? 

Reviewer #1: Yes

Reviewer #2: No

Reviewer #3: N/A

3. Have the authors made all data underlying the findings in their manuscript fully available?

Reviewer #1: Yes

Reviewer #2: Yes

Reviewer #3: Yes

4. Is the manuscript presented in an intelligible fashion and written in standard English?

Reviewer #1: Yes

Reviewer #2: Yes

Reviewer #3: Yes

5. Review Comments to the Author

Reviewer #1: Comments

1. The revised introduction should state the contribution to the literature.

2. Three authors independently conducted the search for relevant research. Is the use of three authors a standard procedure in the literature?

3. Could the studies be weighted by the quality of research design (e.g. cross-sectional vs. panel data estimations)?

4. Six studies in the analysis focus on Germany (page 7). This is a large proportion compared to the economic size of German. Are the results that are presented in the paper globally representative?

5. What was the criteria that the study evaluated “causality”?

6. Older individuals who work and are not retired are those who are the healthiest and probably also the most motivated to work. This may cause potential problems when comparing the results from different countries, because the health status of workers approaching the retirement age is not identical in all countries. For example, the health status of workers maybe better in the countries with universal provision of health care services.

7. The effects can be different in different institutional settings. Is this point relevant for the analysis?

8. The paper should note that there is earlier research that examines the role of job characteristics using linked survey and register data. https://doi.org/10.1016/j.jeoa.2019.02.001

9. The concluding section should state more practical policy conclusions.

Reviewer #2: Report on the paper: “A systematic review of causes of recent increases in ages of labor market exit in OECD countries”

Summary and overall appraisal

According to the authors, the aim of the paper is, on the one hand, to provide an overview of the retirement domains into which current evidence falls, thus identifying research gaps; and, on the other hand, to assess the proportion of change in ages of labor market exit attributable to each investigated cause.

The paper does not have a high technical complexity from a statistical point of view. In fact they do not carry out any statistical analysis by themselves and their conclusions rely on the statistical analysis (mainly econometric analysis) from other research works.

Although the topic studied is relevant and hence worthy of investigation, currently the paper lacks some important issues that should be addressed in a future version of it, in order to reach a publishable standard, from my point of view. Thus, I encourage the authors to take into account seriously the comments detailed below.

Comments

1. As the own authors point out, one of their basic goals is to perform a “systematic review”. In this vein, I miss some relevant references on this literature, for example:

Pérez, C., Martín-Román, Á., & Moral, A. (2015). The impact of leisure complementarity on the labour force participation of older males in Spain. Applied Economics Letters, 22(3), 214-217.

The authors should also check some of the references contained in it. Another interesting paper for a country not belonging to the OECD, but containing literature on this topic is:

Queiroz, B. L., & Souza, L. R. (2017). Retirement incentives and couple’s retirement decisions in Brazil. The Journal of the Economics of Ageing, 9, 1-13.

2. In addition, there is a fresh paper investigating thoroughly one of the domains that the authors identify. Despite being published after the bibliographical database was collected (February 2019), I think it could be also included in a revised version of the paper:

Pérez, C., Martín-Román, Á., & Moral, A. (2019). Two decades of the complementary leisure effect in Spain. The Journal of the Economics of Ageing, forthcoming.

3. Tables 4 and 5 constitute the core of the paper. The authors carry out a descriptive study by reviewing some of the literature on the topic. However, I am afraid that they are comparing a set of very heterogeneous results. I think no clear conclusion can be obtained by just looking at these numbers without a quantitative treatment of them. Particularly, I deem that with such analysis they cannot “assess the proportion of change in ages of labor market exit attributable to each investigated cause” as they state when defining their goals.

4. Following the argument elaborated in my previous comment, I would suggest the authors to perform a meta-analysis. The information contained in tables 4 and 5 might be a promising starting point to do that.

5. I would also suggest the authors to review the writing and to rewrite some parts of paper in a more structured manner.

Reviewer #3: Summary: 

Overall, the paper presents a rough meta-analysis of papers addressing causes of labor-market decisions at older ages. The paper is well-written and was an interesting read. The paper also makes a noble attempt at summarizing the state of knowledge in an extremely important and policy-relevant area. However, I identified a few concerns with the paper.

Concerns: 

1. I am concerned that Column 7 in Table 4 is not recording the "change attributable to explanatory variables" but rather effect sizes. An inspection of the Staublil and Zewimuller (2013) paper seems to just be indicating coefficients rather than changes explained by (the 9.8 and 11.0 are the coefficients of regressions estimated by the paper, not the fraction of change attributable to the explanatory variable). The same seems true of the Qi et al (2018) paper, based on a quick look at an early version of that paper. If this is true, then I'm not sure what the difference is in the papers in Table 4 vs. those in Table 5.

2. I'm not sure if there are other similar meta-analyses done that attempt explicitly what the paper is doing (summarizing literature and addressing gaps). If there are any, they should be mentioned. If not, the authors should make this explicit. 

3. The authors seem to be interested in addressing impacts on retirement ages over time as noted on line 84 and on lines 122-124, to the exclusion of cross-sectional studies. That the authors seem to be interested only in studies with a longitudinal dimension seems a bit odd, since I would assume that some external validity concerns are common to almost all studies. Why are cross-sectional findings not relevant? Is this concern about econometrics or something else? To put things simply, I find it hard to justify ruling-out cross-sectional studies since they can contain equally-valid causal claims about the causes of labor-market choices, so I think the rationale for this needs to be more explicit.  

Much more minor issues: 

3. The discussion on lines 49 through 57 seems a bit awkwardly framed. The authors seem to imply that labor force exit ages are rising, however yet also address issues with what would happen if these ages didn't rise. The discussion could be more consistently framed in a way that raises a need for the findings of the paper.

6. PLOS authors have the option to publish the peer review history of their article (what does this mean?). If published, this will include your full peer review and any attached files.

Reviewer #1: No

Reviewer #2: No

Reviewer #3: No

---

## [Author Response · Author response to Decision Letter 0]

30 Jan 2020

Editor

1. The topic is relevant and hence worthy of investigation, currently the paper lacks some important issues that should be addressed in a future version 

We would like to thank the editor and three anonymous reviewers for their constructive comments. Below, we respond to each of the issues that were raised by the reviewers. Changes in the manuscript are highlighted. 

2. The reviewers did not consider the review systematic since only 18 papers were reviewed and there is a large body of the literature on the topic that should be included. Also, it would be important to developed a more quantitative analysis. One strong suggestion was to use a meta-analysis to study the problem. 

We would like to address separately: (1) the issue of the number of articles included in our review; and (2) the importance of developing a more quantitative analysis. 

(1) Number of articles

Systematic review is a method that ‘seeks to systematically search for, appraise and synthesize research evidence, often adhering to the guidelines on the conduct of a review’ (Grant and Booth 2009, p. 102). The Prisma protocol, which we followed for this review, does not specify a lower boundary for the number of studies that should be included in a systematic review, and neither do other protocols that we are aware of. Other systematic reviews related to retirement reviewed a similar number of studies (e.g., Scharn et al. 2018 reviewed 20; Barnett et al. reviewed 19) or even fewer (Cloostermans et al. reviewed 4). Perhaps, in our original submission, we did not clearly enough describe the goal of our review and the inclusion criteria for the systematic review. Our review is strictly focused on empirical articles that offered an explanation of changes in labor force participation / ages of labor market exit in OECD countries, published since the year 2000. We purposely excluded studies that concentrated on variation in retirement behavior among individuals and between countries at one point in time. These have been summarized in other review articles that we refer to extensively (i.e., Fischer et al. 2017; Scharn et al. 2019). In the revised manuscript, we have modified the introduction section to state the goal of our systematic review more clearly. We have also improved the description of our inclusion criteria to be more explicit. Finally, we assessed the articles that were proposed by the reviewers, and deemed one study relevant for inclusion in our review (i.e. Pérez, Martín-Román & Moral 2019).

(2) Importance of developing a more quantitative analysis

A meta-analysis consists of statistically combining “the results of quantitative studies to provide a more precise effect of the results” (Grant and Booth 2009, p. 98). We agree that a more quantitative analysis would be helpful and we considered it at the earlier stages of work. Unfortunately, a meta-analysis is not an option for this systematic review. The main reason for this is that the methodologies and results presented in our review are, as one reviewer correctly points out, relatively disparate. There exists ample discussion about the kind of results that a meta-analysis should include, but the mainstream knowledge is to only combine studies that include comparable results. Considering the literature that did perform such analysis, narrowly defined dependent and independent variables are usually included. In our case, independent variables denoted considerably varying realities, e.g. change in educational attainment, change in the proportion of workers who contribute to defined contribution schemes, or change in the normal retirement age. Furthermore, studies included in our review had various designs, for example some considered married men ages 55 to 64 in the 1990s in the United States, while others considered women ages 62 to 64 in the 2000s in Switzerland. Methodologies varied considerably too, with some studies using probit models and others relying on a shift-share analysis, for example. As a result, a meta-analysis was not feasible for this article. 

We nevertheless made substantial efforts in order to make the results more comparable across studies. First, quantitative results were originally presented in tables, alongside the context in which they were arrived at. These quantitative results are now presented in graphs, which facilitate comparison between studies. Second, results that were originally presented in Table 4 referred to periods of varying lengths, making comparisons across studies difficult. In the new version, results were converted into annual change, thus again improving comparison between studies. Third, results from Table 5 referred to different metrics but no clear distinction was made between studies according to the metric used. The new version presents results from different studies by grouping them according to the way that the outcome variable was measured, which make comparisons between studies with similar outcomes easier. Fourth, results were further grouped according to classes of predictors in order to improve comparisons. New tables were made which accompany the figures and give important information about the context in which studies were performed. Text in the methods and results sections was adjusted to reflect these changes. In sum, although a meta-analysis appeared impossible to perform, we believe that results in the new manuscript can be more easily compared across studies. 

Furthermore, we reformulated the objective of the review stated in the introduction to reflect better the analyses that are reported in the results section. The way that the second goal of the study is formulated was slightly changed and now reads “Secondly, we quantitatively assess the amount of increases in ages of labor market exit that can be attributed to each investigated cause. Effects are presented individually for each cause, referring to the context in which they were estimated.” 

Finally, we have added a separate limitation section to the discussion, where we discuss how designs and methodologies of the articles that we reviewed vary considerably, warranting caution when interpreting the results. 

3. Introduction should state more clearly the contribution of the paper

We have extensively revised the introduction so that it now states the contribution of the paper more clearly (paragraph 5 in the introduction). 

4. The concluding section should state more practical policy conclusion. 

We have revised the discussion and conclusion sections to clarify the implications for policy makers. 

5. The conclusions and discussion are based on a set of few heterogenous studies. Thus, some conclusions are reached only by looking at results and it is necessary to perform a more quantitative study. 

The issue of performing a more quantitative analysis was addressed above, under comment #2. 

6. it is hard to justify ruling-out cross-sectional studies since they can contain equally-valid causal claims about the causes of labor-market choices, so I think the rationale for this needs to be more explicit. 

We did not a priori rule out cross-sectional studies from our systematic review. However, we explicitly focus on studies that explain changes in ages of labor force exit over time. Conceptually, cross-sectional studies could have been included in our review if they satisfied the inclusion criteria, but none did. Nevertheless, we follow the reviewer’s advice and revised the Methods section to clarify our inclusion criteria, also mentioning that we did not rule out cross-sectional studies. 

7. The paper should note that there is earlier research that examines the role of job characteristics using linked survey and register data. 

The potential inclusion of this research was assessed following our initially established criteria. However, it did not satisfy the criterion stating that studies should aim at explaining change over time in retirement behavior (see also comments under comment #2). 

8. paper needs some revision - text and structure 

The paper underwent a thorough revision of its text and structure. 

9. Please, see more detailed comments below.

The following addresses each detailed comment separately. 

 

Reviewer #1 

1. The revised introduction should state the contribution to the literature.

We have extensively revised the introduction section so that it now states the contribution of this systematic review more clearly (paragraph 5 in the introduction). 

2. Three authors independently conducted the search for relevant research. Is the use of three authors a standard procedure in the literature?

While no strict guidelines are formulated in the PRISMA protocol, a survey of several systematic reviews confirms that the use of three authors is a standard procedure for articles search and inclusion. For example, in Scharn et al. (2018), two authors independently screened titles and abstract, and a third one was consulted if consensus was not reached.

3. Could the studies be weighted by the quality of research design (e.g. cross-sectional vs. panel data estimations)?

All included studies are longitudinal (i.e. panel data), so we cannot compare them with cross-sectional findings. In the new version we specify in the Methods part that studies with either cross-sectional or panel data were considered for inclusion. We chose not to weight studies on any other criteria, as any such criterion and the weight would be relatively arbitrary and add unnecessary complexity. 

4. Six studies in the analysis focus on Germany (page 7). This is a large proportion compared to the economic size of German. Are the results that are presented in the paper globally representative?

Germany, as well as the United States are indeed overrepresented in our results. We have added text to the discussion section (under limitations) to emphasize this issue. While single study findings cannot directly be applied to other countries, we think that this review has considerable value as it is the first to summarize available evidence from different country contexts. 

5. What was the criteria that the study evaluated “causality”?

We have revised the Methods section to clarify our criteria for determining whether a study made efforts towards tracking causality: “Articles were deemed as having made effort towards tracking causality if they used statistical models controlling for potentially spurious correlation and included instruments controlling for endogenous relations, for example by using control groups not affected by the independent variable of interest.”

6. Older individuals who work and are not retired are those who are the healthiest and probably also the most motivated to work. This may cause potential problems when comparing the results from different countries, because the health status of workers approaching the retirement age is not identical in all countries. For example, the health status of workers maybe better in the countries with universal provision of health care services.

We agree with this point, which addresses that the included studies use different populations and refer to different institutional contexts. On the one hand, this is a limitation and we include it in the revised discussion section, under limitations. On the other hand, the goal of this study was to provide an overview of finding from different settings, even if full comparability is impossible. 

7. The effects can be different in different institutional settings. Is this point relevant for the analysis?

This point is related to the previous one. The revised discussion section now mentions this as a limitation. 

8. The paper should note that there is earlier research that examines the role of job characteristics using linked survey and register data. 

The potential inclusion of this research was assessed following our initially established criteria, but it unfortunately did not satisfy the one stating that studies should aim at explaining change over time in retirement behavior (see also editor’s comments under point #2). 

9. The concluding section should state more practical policy conclusions.

 We have revised the discussion and conclusion sections to clarify the implications for policy makers.

 

Reviewer #2

Summary and overall appraisal

According to the authors, the aim of the paper is, on the one hand, to provide an overview of the retirement domains into which current evidence falls, thus identifying research gaps; and, on the other hand, to assess the proportion of change in ages of labor market exit attributable to each investigated cause.

The paper does not have a high technical complexity from a statistical point of view. In fact they do not carry out any statistical analysis by themselves and their conclusions rely on the statistical analysis (mainly econometric analysis) from other research works.

Although the topic studied is relevant and hence worthy of investigation, currently the paper lacks some important issues that should be addressed in a future version of it, in order to reach a publishable standard, from my point of view. Thus, I encourage the authors to take into account seriously the comments detailed below.

Comments

1. As the own authors point out, one of their basic goals is to perform a “systematic review”. In this vein, I miss some relevant references on this literature, for example:

Pérez, C., Martín-Román, Á., & Moral, A. (2015). The impact of leisure complementarity on the labour force participation of older males in Spain. Applied Economics Letters, 22(3), 214-217.

The authors should also check some of the references contained in it. Another interesting paper for a country not belonging to the OECD, but containing literature on this topic is:

Queiroz, B. L., & Souza, L. R. (2017). Retirement incentives and couple’s retirement decisions in Brazil. The Journal of the Economics of Ageing, 9, 1-13.

We have revised the manuscript so that our inclusion criteria are clarified in the introduction. The articles mentioned in this comment and their references were also checked using these criteria, but did not qualify for inclusion in our systematic review (see also reply to editor’s comment #2).

2. In addition, there is a fresh paper investigating thoroughly one of the domains that the authors identify. Despite being published after the bibliographical database was collected (February 2019), I think it could be also included in a revised version of the paper: 

Pérez, C., Martín-Román, Á., & Moral, A. (2019). Two decades of the complementary leisure effect in Spain. The Journal of the Economics of Ageing, forthcoming.

Thank you for this suggestion. This article satisfied our criteria and was therefore included in the revised manuscript. It was probably published after our initial search, which is why it was not included in the previous version. 

3. Tables 4 and 5 constitute the core of the paper. The authors carry out a descriptive study by reviewing some of the literature on the topic. However, I am afraid that they are comparing a set of very heterogeneous results. I think no clear conclusion can be obtained by just looking at these numbers without a quantitative treatment of them. Particularly, I deem that with such analysis they cannot “assess the proportion of change in ages of labor market exit attributable to each investigated cause” as they state when defining their goals.

4. Following the argument elaborated in my previous comment, I would suggest the authors to perform a meta-analysis. The information contained in tables 4 and 5 might be a promising starting point to do that.

We address points 3 and 4 jointly as they are very similar. Please see our reply to the editor’s comment #2. To summarize, we did not perform a meta-analysis because the results are not sufficiently comparable. However, we transformed some of the results into more comparable metrics and improved their presentation to make them more comparable. 

5. I would also suggest the authors to review the writing and to rewrite some parts of paper in a more structured manner.

We have thoroughly revised and rewrote parts of the paper in a more structured manner. 

 

Reviewer #3

Overall, the paper presents a rough meta-analysis of papers addressing causes of labor-market decisions at older ages. The paper is well-written and was an interesting read. The paper also makes a noble attempt at summarizing the state of knowledge in an extremely important and policy-relevant area. However, I identified a few concerns with the paper.

Concerns: 

1. I am concerned that Column 7 in Table 4 is not recording the "change attributable to explanatory variables" but rather effect sizes. An inspection of the Staublil and Zewimuller (2013) paper seems to just be indicating coefficients rather than changes explained by (the 9.8 and 11.0 are the coefficients of regressions estimated by the paper, not the fraction of change attributable to the explanatory variable). The same seems true of the Qi et al (2018) paper, based on a quick look at an early version of that paper. If this is true, then I'm not sure what the difference is in the papers in Table 4 vs. those in Table 5.

We extensively revised our presentation of the results (new tables 4-5 and figures 3-4). The terminology used was checked thoroughly to reflect the actual measures. Tables 4-5 in the revised manuscript may now refer to different studies than in the original manuscript. More specifically, Table 4 now summarizes articles studying differences in LFP between two points in time, while Table 5 summarizes articles studying differences in LFP/ERA between age- or cohort-groups, distinguishing between different types of outcome variable. We have further revised the results and discussion sections to reflect this different approach.

2. I'm not sure if there are other similar meta-analyses done that attempt explicitly what the paper is doing (summarizing literature and addressing gaps). If there are any, they should be mentioned. If not, the authors should make this explicit. 

We are not aware of a systematic review with same aims as ours. We now mention this explicitly in the fifth paragraph of the introduction. 

3. The authors seem to be interested in addressing impacts on retirement ages over time as noted on line 84 and on lines 122-124, to the exclusion of cross-sectional studies. That the authors seem to be interested only in studies with a longitudinal dimension seems a bit odd, since I would assume that some external validity concerns are common to almost all studies. Why are cross-sectional findings not relevant? Is this concern about econometrics or something else? To put things simply, I find it hard to justify ruling-out cross-sectional studies since they can contain equally-valid causal claims about the causes of labor-market choices, so I think the rationale for this needs to be more explicit. 

We agree that longitudinal and cross-sectional studies can contain equally valid causal claims about the causes of labor market choices. As mentioned above (editor’s comment #6), we did not a priori rule out cross-sectional studies. It is important to understand, however, that our focus was not on labor market choices in general, but on change over time in ages of labor force exit. Conceptually, cross-sectional studies could have been included in our review if they satisfied this criterion, but our search did not yield any such study. We have revised the Methods section to clarify our inclusion criteria. It also mentions that cross-sectional studies were not explicitly ruled out. 

Much more minor issues: 

4. The discussion on lines 49 through 57 seems a bit awkwardly framed. The authors seem to imply that labor force exit ages are rising, however yet also address issues with what would happen if these ages didn't rise. The discussion could be more consistently framed in a way that raises a need for the findings of the paper.

We agree with this point and have thoroughly revised this part of the discussion section, emphasizing the findings of the paper. 

 

References

Barnett, I., van Sluijs, E. M., & Ogilvie, D. (2012). Physical activity and transitioning to retirement: a systematic review. American journal of preventive medicine, 43(3), 329-336.

Cloostermans, L., Bekkers, M. B., Uiters, E., & Proper, K. I. (2015). The effectiveness of interventions for ageing workers on (early) retirement, work ability and productivity: a systematic review. International archives of occupational and environmental health, 88(5), 521-532.

Fisher, G. G., Chaffee, D. S., & Sonnega, A. (2016). Retirement timing: A review and recommendations for future research. Work, Aging and Retirement, 2(2), 230-261.

Grant, M. J., & Booth, A. (2009). A typology of reviews: an analysis of 14 review types and associated methodologies. Health Information & Libraries Journal, 26(2), 91-108.

Pérez, C., Martín-Román, Á., & Moral, A. (2020). Two decades of the complementary leisure effect in Spain. The Journal of the Economics of Ageing, 15, 100216.

Scharn, M., Sewdas, R., Boot, C. R., Huisman, M., Lindeboom, M., & Van Der Beek, A. J. (2018). Domains and determinants of retirement timing: A systematic review of longitudinal studies. BMC public health, 18(1), 1083.

---

## [Decision Letter · Decision Letter 1]

11 Mar 2020

PONE-D-19-28928R1

A systematic review of causes of recent increases in ages of labor market exit in OECD countries

PLOS ONE

Dear Dr. Boissonneault,

Thank you for submitting your manuscript to PLOS ONE. After careful consideration, we feel that it has merit but does not fully meet PLOS ONE’s publication criteria as it currently stands. Therefore, we invite you to submit a revised version of the manuscript that addresses the points raised during the review process.

We appreciate the authors’ receptiveness to earlier critiques and their inclusion of changes and the authors’ receptiveness to earlier critiques and their inclusion of changes. However, based on the reviewers comments and my own reading, I believe the paper needs some additional adjustments. 

 In general, there is an disagrement by how  systematic is the review. There are some  miussing relevant references on this literature.the descriptive analysis do not lead to the conclude that the paper  assess “the proportion of change in ages of labor market exit attributable to each investigated cause”,one suggestions is to incorporate a broader literature and perform a meta-analysisthe paper does not carry out any statistical analysis but, the conclusions rely on the statistical analysis. So, we believe this is an important adjustment to the paper.The detailed comments by Reviewer #2 and # 3 are organized in the attached document. 

We would appreciate receiving your revised manuscript by Apr 25 2020 11:59PM. To enhance the reproducibility of your results, we recommend that if applicable you deposit your laboratory protocols in protocols.io, where a protocol can be assigned its own identifier (DOI) such that it can be cited independently in the future. For instructions see: http://journals.plos.org/plosone/s/submission-guidelines#loc-laboratory-protocols

We look forward to receiving your revised manuscript.

Kind regards,

Bernardo Lanza Queiroz, Ph.D

Academic Editor

PLOS ONE

Reviewers' comments:

Reviewer's Responses to Questions

**Comments to the Author**

1. If the authors have adequately addressed your comments raised in a previous round of review and you feel that this manuscript is now acceptable for publication, you may indicate that here to bypass the “Comments to the Author” section, enter your conflict of interest statement in the “Confidential to Editor” section, and submit your "Accept" recommendation.

Reviewer #1: All comments have been addressed

Reviewer #2: (No Response)

Reviewer #3: (No Response)

2. Is the manuscript technically sound, and do the data support the conclusions?

Reviewer #1: Yes

Reviewer #2: No

Reviewer #3: Partly

3. Has the statistical analysis been performed appropriately and rigorously? 

Reviewer #1: Yes

Reviewer #2: No

Reviewer #3: No

4. Have the authors made all data underlying the findings in their manuscript fully available?

Reviewer #1: No

Reviewer #2: Yes

Reviewer #3: Yes

5. Is the manuscript presented in an intelligible fashion and written in standard English?

Reviewer #1: Yes

Reviewer #2: Yes

Reviewer #3: Yes

6. Review Comments to the Author

Reviewer #1: I am happy with the revised version of the paper. I like the research question, the structure of the paper, the quality of writing, and the way the authors describe their empirical proceeding and results. Most importantly, the authors have addressed all the issues stated in my referee report for the first version appropriately.

Reviewer #2: Report on the paper: “A systematic review of causes of recent increases in ages of labor market exit in OECD countries” Revision 1.

First of all, I would like to acknowledge that the authors have carried out an important review work. Thus, I would like to congratulate them for that. I strongly believe that the paper has improved significantly in this second round. However, I still have some important remarks to make.

1. Now I understand the label “systematic review”. The authors explain that it is the name of a research technique. In any case, the number of articles used is still very small to draw any sensible “general” conclusion. Despite the fact that they make use of arguments from authority to justify this sample size, I cannot see how they can make any inference for the OECD countries. Particularly with such an unbalanced sample of countries (e.g. six studies for Germany and so on). I align myself here with reviewer 1.

2. As for the quantitative analysis, the authors claim that “a meta-analysis was not feasible for this article”. I do not agree with that. It is true, however, that they should define an appropriate dependent variable (this the first and key decision to be made) and a set of independent variables, many of them defined as dummy variables. In order to discard the meta-analysis, they state that “their independent variables denoted varying realities”, “studies included had various designs” and “methodologies varied considerably too”. I wonder if such “flaws” do not impede to perform a wise qualitative analysis. My point of view is different: a good meta-analysis could overcome that issues.

3. As regards the quantitative analysis again, the authors allege to have performed some quantitative analysis. I do not think so. For instance, they state: “quantitative results were originally presented in tables”. In my view, what they call quantitative results are only some figures extracted from other research works. I do not want to appear as a maximalist quantitative researcher, but the truth is that PLOS ONE is a journal that specifically encourages the use of a sound statistical analysis. In fact, one of the questions I have to answer as a reviewer is: “Has the statistical analysis been performed appropriately and rigorously?” To be honest there is a lack of statistical analysis. There is only a use of statistics from a descriptive perspective.

4. After reading the authors’ response to my first comment in the previous round, I am seriously questioning the inclusion criteria followed. If Perez et al. (2015) does not qualify for inclusion, that criteria should be revised. I affirm this because this article practically mimics the work by Schirle (2008), which is included.

Reviewer #3: I would supplement the data included in the paper with exactly what point estimates and what results were used from each paper.

7. PLOS authors have the option to publish the peer review history of their article (what does this mean?). If published, this will include your full peer review and any attached files.

Reviewer #1: No

Reviewer #2: No

Reviewer #3: No

---

## [Author Response · Author response to Decision Letter 1]

20 Mar 2020

See attached document for a complete response to all comments from the three reviewers and the editor.

---

## [Editor Report · Decision Letter 2]

3 Apr 2020

A systematic review of causes of recent increases in ages of labor market exit in OECD countries

PONE-D-19-28928R2

Dear Dr. Boissonneault,

We are pleased to inform you that your manuscript has been judged scientifically suitable for publication and will be formally accepted for publication once it complies with all outstanding technical requirements.

With kind regards,

Bernardo Lanza Queiroz, Ph.D

Academic Editor

PLOS ONE
---

## [Editor Report · Acceptance letter]

13 Apr 2020

PONE-D-19-28928R2 

A systematic review of causes of recent increases in ages of labor market exit in OECD countries 

Dear Dr. Boissonneault:

I am pleased to inform you that your manuscript has been deemed suitable for publication in PLOS ONE. Congratulations! Your manuscript is now with our production department. 

With kind regards,

on behalf of

Dr. Bernardo Lanza Queiroz 

Academic Editor

PLOS ONE